# Inhibition of Fast Nerve Conduction Produced by Analgesics and Analgesic Adjuvants—Possible Involvement in Pain Alleviation

**DOI:** 10.3390/ph13040062

**Published:** 2020-04-05

**Authors:** Eiichi Kumamoto

**Affiliations:** Department of Physiology, Saga Medical School, 5-1-1 Nabeshima, Saga 849-8501, Japan; kumamote@cc.saga-u.ac.jp

**Keywords:** analgesic, analgesic adjuvant, antinociception, nerve conduction, sciatic nerve, compound action potential, Na^+^ channel, K^+^ channel

## Abstract

Nociceptive information is transmitted from the periphery to the cerebral cortex mainly by action potential (AP) conduction in nerve fibers and chemical transmission at synapses. Although this nociceptive transmission is largely inhibited at synapses by analgesics and their adjuvants, it is possible that the antinociceptive drugs inhibit nerve AP conduction, contributing to their antinociceptive effects. Many of the drugs are reported to inhibit the nerve conduction of AP and voltage-gated Na^+^ and K^+^ channels involved in its production. Compound action potential (CAP) is a useful measure to know whether drugs act on nerve AP conduction. Clinically-used analgesics and analgesic adjuvants (opioids, non-steroidal anti-inflammatory drugs, α_2_-adrenoceptor agonists, antiepileptics, antidepressants and local anesthetics) were found to inhibit fast-conducting CAPs recorded from the frog sciatic nerve by using the air-gap method. Similar actions were produced by antinociceptive plant-derived chemicals. Their inhibitory actions depended on the concentrations and chemical structures of the drugs. This review article will mention the inhibitory actions of the antinociceptive compounds on CAPs in frog and mammalian peripheral (particularly, sciatic) nerves and on voltage-gated Na^+^ and K^+^ channels involved in AP production. Nerve AP conduction inhibition produced by analgesics and analgesic adjuvants is suggested to contribute to at least a part of their antinociceptive effects.

## 1. Introduction

Nociceptive information from the periphery to the cerebral cortex is mainly transmitted by action potential (AP) conduction in nerve fibers and chemical transmission at synapses (see [1,2] for reviews). Nociceptive pain is usually acute and relieved by narcotic analgesics such as opioids and antipyretic analgesics including non-steroidal anti-inflammatory drugs (NSAIDs). On the other hand, pain may persist or recur for longer than three months, which is called chronic pain. One type of chronic pain, neuropathic pain, which occurs as a result of damage to the peripheral or central nervous system (PNS and CNS, respectively), is characterized by a hyper-excitability of neurons near the injured neuronal tissues (see [3] for review). This type of pain is often resistant to analgesics such as opioids and NSAIDs and is thus treated by using analgesic adjuvants such as α_2_-adrenoceptor agonists, antiepileptics, antidepressants and local anesthetics (see [4,5,6,7,8,9,10,11] for reviews). Although the main target of analgesics and analgesic adjuvants, except for NSAIDs and local anesthetics, is generally synapses (see [12,13,14] for reviews), it is possible that all of their drugs inhibit nerve AP conduction, partly contributing to their inhibitory effects on pain.

The AP conduction is mediated by voltage-gated Na^+^ and K^+^ channels located in nerve fibers. Thus, a depolarizing stimulus given to a nerve fiber point activates Na^+^ channels expressed in membranes of the fiber, allowing Na^+^ entry to the cytoplasm, caused by the gradient of the electrochemical potential of Na^+^, leading to a self-regenerative production of AP. This in turn results in an outward current (membrane depolarization) in a fiber point adjacent to the point to produce opening of other Na^+^ channels and so forth. Such a production of AP subsides by a subsequent inactivation of Na^+^ channels and activation of K^+^ channels (see [15,16] for reviews).

In a bundle of nerve fibers exposed to insulator such as oil, sucrose or air, AP conduction in each fiber produces AP current flowing through nerve bundle surface having high resistance that can be measured as a potential difference, i.e., compound action potential (CAP), by using two electrodes put on the nerve. Sciatic nerve trunk dissected from frogs is a useful preparation to easily and stably record voltage-gated Na^+^-channel blocker tetrodotoxin (TTX)-sensitive and fast-conducting (possibly myelinated Aα-fiber mediated) CAPs by using the nerve trunk exposed to air (air-gap method). A voltage-gated delayed-rectifier K^+^-channel inhibitor tetraethylammonium increased the half-peak duration of the CAP with no change in its peak amplitude, indicating an involvement of K^+^ channels (see [17] for review). Although not only fast-conducting but also slow-conducting (C-fiber mediated) CAPs were recorded from the frog sciatic nerve, the latter had much smaller peak amplitude and conduction velocity than the former [18].

Frog sciatic nerve fast-conducting CAPs were found to be inhibited by clinically-used antinociceptive drugs such as many kinds of opioids including tramadol [19,20], NSAIDs [21], an α_2_-adrenoceptor agonist dexmedetomidine ((+)-(*S*)-4-[1-(2,3-dimethylphenyl)-ethyl]-1*H*-imidazole or DEX [22]), antiepileptics [23], antidepressants [24] and many kinds of local anesthetics (lidocaine, ropivacaine, prilocaine, levobupivacaine, bupivacaine, cocaine, procaine, benzocaine, tetracaine and pramoxine [19,20,23,25,26,27]) and also by a general anesthetic propofol [25]. These inhibitory actions were concentration-dependent and depended on the chemical structures of the drugs. This review article will mention the effects of the analgesics and analgesic adjuvants on CAPs recorded from the frog sciatic nerve and discuss a difference in nerve AP conduction inhibition among the drugs. For comparison, it will be described how the antinociceptive drugs affect mammalian peripheral nerve CAPs and voltage-gated Na^+^ and K^+^ channels involved in the production of AP, when data are available. Moreover, it will be shown that antinociceptive plant-derived compounds also inhibit frog sciatic nerve CAPs with efficacies comparable to those of NSAIDs and analgesic adjuvants.

## 2. Actions of Analgesics on Nerve Conduction

### 2.1. Opioids

Opioids are well-known to inhibit glutamatergic excitatory transmission by activating opioid receptors in the CNS including the central terminals of primary-afferent fibers, resulting in antinociception ([28,29,30]; see [31,32] for reviews). Not only central but also peripheral terminal opioid receptors in primary-afferent neurons are thought to be involved in antinociception ([33,34,35,36]; see [37] for review). Opioids also exhibit a local anesthetic effect in the PNS. Although it has been reported in decerebrated cats that the perineural administration of an opioid morphine has no effect on CAPs in the superficial radial nerve [38], AP conduction in peripheral nerve fibers is generally blocked by opioids. For example, opioids such as fentanyl and sufentanil decreased the peak amplitudes of CAPs recorded from peripheral nerve fibers [39] and inhibited peripheral nerve AP conduction [40].

A morphine-induced CAP inhibition in mammalian peripheral nerve fibers was antagonized by a non-specific opioid-receptor antagonist naloxone, indicating an involvement of opioid receptors [41]. Consistent with this observation, binding and immunohistochemical studies have shown the localization of opioid receptors in mammalian peripheral nerve fibers [42,43,44]. It has been also demonstrated that a frog sciatic nerve CAP inhibition produced by opioids is sensitive to naloxone [45]. On the contrary, there are reports showing that opioids decrease CAP peak amplitudes [39] and suppress nerve AP conduction [40] in a manner resistant to naloxone.

#### 2.1.1. Tramadol

The compound (1*RS*,2*RS*)-2-[(dimethylamino)methyl]-1-(3-methoxyphenyl)cyclohexanol hydrochloride (tramadol) is an orally-active opioid which is clinically used as an analgesic in the CNS [46]. Although tramadol is metabolized to various compounds such as mono-*O*-desmethyltramadol (M1) via *N*- and *O*-demethylation in animals and humans [47], M1 is a therapeutically active drug as a central analgesic [46]. One of cellular mechanisms for the antinociceptive effect of tramadol is the activation of μ-opioid receptors [48,49]. In agreement with this idea, M1 has the highest affinity for cloned μ-opioid receptors among the metabolites of tramadol. M1 is reported to inhibit glutamatergic excitatory transmission in spinal lamina II (substantia gelatinosa) neurons which play a crucial role in regulating nociceptive transmission to the spinal dorsal horn from the periphery, resulting in a decrease in the excitability of the neurons [50,51,52]. In addition to such a central activity, tramadol is known to have a local anesthetic effect following its intradermal injection in patients ([53,54,55]; see [56] for review). Consistent with this result, in vivo studies have shown a spinal somatosensory evoked potential inhibition produced by a direct application of tramadol to the rat sciatic nerve [57].

Tramadol reduced the peak amplitude of CAPs recorded from the frog sciatic nerve in a concentration-dependent manner in a range of 0.2 to 5 mM [19]. A similar CAP inhibitory action of tramadol has been reported by other investigators in the frog [58] and rat sciatic nerve [59,60]. Analysis based on the Hill equation demonstrated that half-maximal inhibitory concentration (IC_50_) value for tramadol to reduce frog sciatic nerve CAP amplitudes is 2.3 mM; this IC_50_ value was smaller by about three-fold than that (6.6 mM) reported previously for the frog sciatic nerve [58]. Rat sciatic nerve CAPs were inhibited by tramadol (37% peak amplitude reduction at 4 mM) less effectively than frog sciatic nerve’s ones [59]. This inhibitory action of tramadol in the frog sciatic nerve was not affected by the pretreatment of the sciatic nerve with naloxone (0.01 mM); a μ-opioid receptor agonist (D-Ala^2^, *N*-Me-Phe^4^, Gly^5^-ol)enkephalin (DAMGO; 1 μM) had no effect on frog sciatic nerve CAPs. Furthermore, CAPs were affected by much smaller extents by M1 (see below) that is similar in chemical structure to tramadol while exhibiting a higher affinity for μ-opioid receptors than tramadol [61]. These results indicate that the tramadol-induced CAP inhibition is not mediated by opioid receptors [19]. Consistent with this result, a spinal somatosensory evoked potential inhibition following the application of tramadol on rat sciatic nerves in vivo was resistant to naloxone [57]. Although tramadol inhibits noradrenaline (NA) and serotonin (5-hydroxytryptamine; 5-HT) reuptake at concentrations similar to those that activate μ-opioid receptors [62,63], a combination of inhibitors of the reuptake of NA and 5-HT (desipramine and fluoxetine, respectively; each 10 μM; see Section 3.3) did not affect frog sciatic nerve CAPs, indicating no involvement of NA and 5-HT reuptake inhibition in the CAP inhibition [19].

The CAP inhibition produced by tramadol is possibly due to an inhibition of voltage-gated Na^+^ and K^+^ channels involved in the production of AP. Tramadol concentration-dependently reduced the peak amplitude of TTX-sensitive Na^+^ channel currents recorded from dorsal root ganglion (DRG) neuroblastoma hybridoma cell line ND7/23 cells with an IC_50_ value of 0.194 mM [64] and from HEK293 cells expressing rat Nav1.2 channels with an IC_50_ value of 0.103 mM [65]. These values were smaller than that (2.3 mM) for frog sciatic nerve CAP inhibition. It has been demonstrated that tramadol suppresses the current amplitude of delayed rectifier K^+^-channels (Kv3.1a type) expressed in NG 108-15 cells with an IC_50_ of 0.025 mM, a value much less than 2.3 mM [66]. Such IC_50_ values for tramadol to inhibit CAPs, Na^+^ and K^+^ channels were higher than its clinically relevant concentration of about 2 μM in serum [52,67].

Unlike tramadol, M1 (1-2 mM) did not affect frog sciatic nerve CAPs. This was confirmed in the frog sciatic nerve whose CAPs were inhibited by tramadol (1 mM; [19]). CAP peak amplitudes were reduced by 9% by M1 at 5 mM. Consistent with such smaller effects of M1, APs conducting on rat primary-afferent fibers were not blocked when the effect of M1 (1 mM) on dorsal root-evoked excitatory postsynaptic currents was examined by applying the patch-clamp technique to lamina II neurons in spinal cord slices [51]. It is interesting to note that tramadol has -OCH_3_ bound to the benzene ring while M1 has -OH and thus that the methyl group is present in tramadol but not M1. This result indicates that the difference in chemical structure between tramadol and M1 is responsible for the distinction in CAP inhibition (see Figure 5a in [19] for a comparison of the chemical structures of the two compounds).

#### 2.1.2. Other Opioids

In order to reveal whether such a structure-activity relationship is seen for other opioids, the effects of morphine, codeine and ethylmorphine on frog sciatic nerve CAPs were examined. Morphine concentration-dependently reduced CAP peak amplitude; this extent at 5 mM was 15%. Codeine, which has -OCH_3_ in place of -OH in morphine, at 5 mM reduced CAP peak amplitude by 30%. Moreover, CAPs were more effectively inhibited by ethylmorphine, where -OH of morphine is replaced by -OCH_2_CH_3_; amplitude reduction at 5 mM was 61%. IC_50_ value for ethylmorphine in reducing CAP peak amplitudes was 4.6 mM. CAP inhibitions produced by morphine (10 mM), codeine (5 mM) and ethylmorphine (2 mM) were resistant to naloxone (0.01 mM). Naloxone at a high concentration such as 1 mM by itself reduced by 9% CAP peak amplitudes, but did not affect morphine (10 mM) activity [20]. These results indicate no involvement of opioid receptors in the CAP inhibition produced by opioids, as reported previously in mammalian peripheral nerves [39,40,68]. A sequence of the opioid-induced CAP peak amplitude reduction was ethylmorphine > codeine > morphine. Thus, CAP amplitude reduction increased in extent with an increase in the number of -CH_2_ (see Figure 7A in [20] for a comparison of the chemical structures of the three opioids). This result is consistent with the above-mentioned observation that tramadol having -OCH_3_ in the benzene ring inhibits CAPs more effectively than M1 which is different from tramadol only in terms of the presence of -OH in the ring. Interestingly, this result was obtained in spite of the fact that the chemical structures of morphine, codeine and ethylmorphine are quite distinct from those of tramadol and M1 (see [69] for review). Since the increase in -CH_2_ number is thought to enhance lipophilicity of opioids, lipophilic opioid-channel interaction is suggested to play a pivotal role in nerve AP conduction block, as shown for local anesthetics [70,71]. This idea is supported by the observation that the potency in CAP inhibition in the rat sciatic nerve was in the order of isopropylcocaine (where the methyl ester group of cocaine is replaced with an isopropyl ester group) > cocaethylene (where its methyl ester group is replaced with an ethyl ester group) > cocaine [72]). It is interesting to note that the sequence of the affinity of opioids for μ-opioid receptors is morphine > codeine > ethylmorphine [73], the order of which is reversed to one for CAP inhibition. If the opioid-induced inhibition of CAPs is mediated by μ-opioid receptors, CAP inhibition sequence will be expected to be morphine > codeine > ethylmorphine. However, this sequence is not seen, a result being consistent with the idea that the opioids-induced frog sciatic nerve CAP inhibition is not mediated by opioid receptors.

The same sequence as that in the frog sciatic nerve has been reported in the rat phrenic nerve [68], although there is a quantitative difference between the two studies. When compared at 5 mM, codeine-induced reduction (about 30%) in frog sciatic nerve CAP peak amplitude was much smaller than that (about 70%) in the rat phrenic nerve, while so a large distinction was not seen in morphine action (about 10%). Frog sciatic nerve CAPs were less sensitive to morphine than those in rabbit and guinea-pig vagus nerves in such that vagus nerve CAP peak amplitudes were reduced by 20–32% at 0.5 mM [41]. APs recorded intracellularly from rat DRG neurons having Aα/β myelinated primary-afferent fibers also exhibited the sequence of ethylmorphine > codeine ≥ morphine (IC_50_ = 0.70, 2.5 and 2.9 mM, respectively) in AP peak amplitude reduction; this inhibition was resistant to naloxone (0.01 mM) [74].

Although many drugs including narcotics, antiepileptics, local anesthetics, alcohols and barbiturates block AP conduction in peripheral nerve fibers, suggesting a nonspecific interaction of the drugs with membrane bilayers [75], the chemical structure-specific CAP inhibition produced by opioids indicates that opioids act on proteins such as voltage-gated Na^+^ and K^+^ channels (see [76] for review). Morphine is reported to suppress peak Na^+^ currents and steady-state K^+^ currents in single myelinated nerve fibers isolated from the frog sciatic nerve, leading to the prolongation of APs [77]. Intracellularly-applied morphine reduced voltage-gated Na^+^ and K^+^ channel current amplitudes in squid giant axons [78]. Bath-applied morphine reduced TTX-sensitive Na^+^ channel current amplitude in DRG neuroblastoma hybridoma cell line ND7/23 cells with an IC_50_ value of 0.378 mM [64], although Nav1.2 channels expressed in HEK293 cells were unaffected by morphine at 1 mM [65]. In support of such an idea about ion channel inhibition, it has been reported that an opioid meperidine, which is used for AP conduction blockade and thus analgesia, inhibits Na^+^-channels in a manner similar to that of lidocaine [79]. Table 1 summarizes IC_50_ values for frog sciatic nerve fast-conducting CAP inhibitions produced by opioids together with those for rat sciatic nerve CAPs and voltage-gated Na^+^ channels.

In clinical practice, although administration of opioids into the nerve sheath results in pain relief (for instance, see [80]), many of pain treatments by use of opioids are due to systemic administration of centrally-penetrating opioids, leading to their actions in the PNS and CNS, both of which contribute to analgesia (see [81] for review). It is possible that centrally-administrated opioids act on not only the CNS but also the PNS, because opioids are reported to be transported to the periphery from brain by P-glycoprotein [82]. In support of an important role of opioids in the PNS, subcutaneous administration of *N*-methyl-morphine, which did not pass through the blood brain barrier, resulted in antinociception in an acetic acid-writhing model in mice [35]. It has been reported that a subcutaneously-administrated opioid loperamide, which cannot penetrate into the brain, exhibited an antinociceptive effect in the formalin test in rats [34]. Such an action of opioids in the PNS appeared to be mediated by opioid receptors in peripheral terminals of primary-afferent fibers ([33,34,35,36,83]; see [37] for review). In addition, the inhibitory effect of opioids on nerve AP conduction also might contribute to local analgesia following the peripheral perineural administration of opioids (for instance, see [84]) that are expected to lead to a direct action of opioids at high doses on peripheral nerves. Since codeine is metabolized to morphine via *O*-demethylation in humans and animals ([85,86]; see [81] for review), peripherally-administrated codeine might have a similar effect to that of morphine.

### 2.2. NSAIDs

Antinociception produced by NSAIDs is mediated by various mechanisms such as (1) inhibition of the synthesis of prostaglandins from arachidonic acid by inhibiting the cyclooxygenase enzyme ([87,88]; see [89,90,91] for reviews), (2) inhibition of acid-sensitive ion channels [92] and transient receptor potential (TRP) channels [93,94], (3) activation of several K^+^ channels ([95,96,97,98,99]; see [100,101] for reviews), (4) substance P depletion [102], (5) an interaction with the adrenergic system [103] and (6) an involvement of opioids [104,105] and endocannabinoids (see [106] for review). The idea about an involvement of mechanisms other than cyclooxygenase inhibition in antinociception is supported by the observation that there is a dissociation between anti-inflammation and antinociception produced by NSAIDs [107].

An acetic acid-based NSAID diclofenac reduced frog sciatic nerve CAP peak amplitudes in a partially reversible manner. Diclofenac activity was concentration-dependent in a range of 0.01–1 mM with an IC_50_ value of 0.94 mM. Another acetic acid-based NSAID aceclofenac (a carboxymethyl ester of diclofenac) also exhibited a similar CAP inhibitory action. CAP peak amplitudes were concentration-dependently reduced by aceclofenac in a range of 0.01–1 mM with an IC_50_ of 0.47 mM, a value smaller than that of diclofenac. Other acetic acid-based NSAIDs had an efficacy smaller than those of diclofenac and aceclofenac. Indomethacin at 1 mM reduced CAP peak amplitudes by 38% and acemetacin (where the -OH group of indomethacin is substituted by -OCH_2_COOH) at 0.5 mM did so by 38%. Etodolac at 1 mM reduced CAP peak amplitudes by only 15%, and sulindac and felbinac at 1 mM had no effects on CAP peak amplitudes [21].

A similar frog sciatic nerve CAP inhibition was produced by fenamic acid-based NSAIDs (tolfenamic acid, meclofenamic acid, mefenamic acid and flufenamic acid) whose chemical structures are similar to those of diclofenac and aceclofenac. Tolfenamic acid concentration-dependently reduced CAP peak amplitudes in a range of 0.01–0.2 mM with an IC_50_ value of 0.29 mM. The activity of meclofenamic acid (where the chloro group bound to the benzene ring of tolfenamic acid is changed in number and position) was concentration-dependent in a range of 0.01–0.5 mM with an IC_50_ value of 0.19 mM. Moreover, mefenamic acid (where the chloro group bound to the benzene ring of tolfenamic acid is replaced by methyl group) concentration-dependently reduced CAP peak amplitudes in a range of 0.01–0.2 mM with the extent of 16% at 0.2 mM. CAP peak amplitudes were concentration-dependently reduced by flufenamic acid (where one out of two methyl groups bound to the benzene ring of mefenamic acid is lacking and another one is replaced by -CF_3_) with an IC_50_ of 0.22 mM, a value comparable to those of tolfenamic acid and meclofenamic acid [21].

2,6-Dichlorodiphenylamine and *N*-phenylanthranilic acid (which are similar in chemical structure to diclofenac and tolfenamic acid while being not NSAIDs) reduced frog sciatic nerve CAP peak amplitudes; the former compound lacks the -CH_2_COOH group of diclofenac and the latter one lacks chloro and methyl groups bound to the benzene ring of tolfenamic acid. 2,6-Dichloro- diphenylamine activity was concentration-dependent in a range of 0.001-0.1 mM with the extent of 45% at 0.1 mM; *N*-phenylanthranilic acid activity was concentration-dependent in a range of 0.01–2 mM with the extent of 23% at 1 mM [21].

With respect to other types of NSAIDs, salicylic acid-based (aspirin; 1 mM), propionic acid-based (ketoprofen, naproxen, ibuprofen, loxoprofen and flurbiprofen; each 1 mM) and enolic acid-based [meloxicam (0.5 mM) and piroxicam (1 mM)] NSAIDs had no effects on frog sciatic nerve CAP amplitudes [21].

CAP amplitude reductions produced by the NSAIDs would be mediated by an inhibition of TTX-sensitive voltage-gated Na^+^ channels that are involved in frog CAP production. In support of this idea, diclofenac decreased the peak amplitudes of TTX-sensitive Na^+^-channel currents in rat DRG [108] and mouse trigeminal ganglion neurons [109]. A similar diclofenac-induced Na^+^-channel inhibition has been reported in rat myoblasts [110] and ventricular cardiomyocytes [111]. Flufenamic acid as well as diclofenac decreased Na^+^-channel current amplitudes in rat hippocampal CA1 neurons [112,113,114]. Although IC_50_ value (0.22 mM) for flufenamic acid in frog sciatic nerve CAP inhibition was similar to that of Na^+^ channel inhibition (0.189 mM) in rat hippocampal CA1 neurons [114], IC_50_ value (0.94 mM) for diclofenac in CAP inhibition was much larger than those (0.00851 and 0.014 mM in rat myoblasts and DRG neurons, respectively) of Na^+^ channel inhibition [108,110]. Regarding rank order among NSAIDs, the order for CAP inhibition at 0.5 mM was flufenamic acid > diclofenac > indomethacin >> aspirin = naproxen = ibuprofen [21]; this was in part similar to those for Na^+^ channel inhibition in rat cardiomyocytes (diclofenac > naproxen ≥ ibuprofen; [111]) and also in rat DRG neurons (diclofenac > flufenamic acid > indomethacin > aspirin; [108]). With respect to TTX-resistant Na^+^ channels, diclofenac at 0.3 mM reduced peak current amplitudes by about 20% in rat trigeminal ganglion neurons [115]; Nav1.8 channel currents were inhibited by flufenamic acid and tolfenamic acid (current amplitude reduction: ca. 30 and 30%, respectively, at 0.1 mM; [116]). TTX-sensitive Nav1.7 channel currents were more sensitive to flufenamic acid and tolfenamic acid (reduction: ca. 60 and 70%, respectively, at 0.1 mM) than Nav1.8 ones [116]. Alternatively, chemical irritation-induced activity increase of cat corneal sensory nerve fibers was suppressed in extent by NSAIDs; this suppression was different in magnitude among distinct types of NSAIDs [109,117]. Na^+^-channel inhibition produced by NSAIDs appeared to be different in extent among preparations. Concentrations required for NSAIDs to have a significant inhibitory effect on frog sciatic nerve CAPs were in general higher than those needed to inhibit Na^+^ channels; this may be attributed to various reasons including the fact that not only Na^+^ channels but also K^+^ channels are involved in determining CAP amplitudes. To my knowledge, the effects of aceclofenac, indomethacin, etodolac, acemetacin, meclofenamic acid and mefenamic acid on voltage-gated Na^+^ channels have not been reported. Table 2 summarizes IC_50_ values for frog sciatic nerve fast-conducting CAP inhibitions produced by NSAIDs together with those for voltage-gated Na^+^ channels.

NSAIDs (diclofenac, aceclofenac, tolfenamic acid, meclofenamic acid and flufenamic acid), which are more effective in frog sciatic nerve CAP inhibition compared to the other NSAIDs [21], have two benzene rings that bind a hydrophilic substituent group, both of which rings are linked by -NH- (see Figures 1Aa, 1Ba, 3Aa, 3Ba, 3Da in [21] for the chemical structures of the five NSAIDs), as seen in local anesthetics (see Section 3.4). Mefenamic acid (where one of the two benzene rings has a hydrophobic substituent group; see Figure 3Ca in [21]) appeared to be less effective, albeit not examined at a higher concentration due to a less solubility of this drug (see above). CAPs were effectively inhibited by 2,6-dichlorodiphenylamine and *N*-phenylanthranilic acid that are not NSAIDs while being similar in chemical structure to NSAIDs having two benzene rings (see Figures 4Aa and 4Ba in [21]). CAPs were also depressed by bisphenol A that have two benzene rings that bind a hydrophilic group such as -OH [26].

Much evidence demonstrates that the other actions of NSAIDs depend on their chemical structures. For instance, an involvement of NO-cGMP-K^+^ channels in antinociception mediated by NSAIDs was dependent on their chemical structures [98,118]. Nonselective cation channels in the rat exocrine pancreas were suppressed by flufenamic acid and mefenamic acid but not indomethacin, aspirin and ibuprofen [119]. There was a distinction in depressing TRP melastatin-3 channels between diclofenac and aceclofenac [94]. Although NSAIDs not only suppress but also activate TRP ankyrin-1 channels, this activation also differed in magnitude among NSAIDs [120]. Moreover, there was a distinction among NSAIDs in the activities of mitochondrial oxidative phosphorylation or electron transport system that may be involved in their adverse side effects [121].

Although the concentrations of NSAIDs tested in the frog sciatic nerve are generally much higher than those for voltage-gated Na^+^-channel inhibition, such high concentrations are likely when NSAIDs are used at high concentrations in the direct vicinity of nerve fibers. At least a part of analgesia caused by NSAIDs used as a dermatological drug for antinociception may be due to a nerve conduction inhibition through their inhibitory action on voltage-gated Na^+^ channels [122].

## 3. Actions of Analgesic Adjuvants on Nerve Conduction

### 3.1. Adrenoceptor Agonists

Intrathecally or epidurally administrated α_2_ agonists such as clonidine and DEX (see [123] for review) produce analgesia in animals [124,125,126] and humans [127]. This is possibly due to inhibited glutamatergic excitatory transmission in spinal superficial dorsal horn neurons [128]. α_2_ Agonists combined with local anesthetics in spinal anesthesia extend the duration of peripheral nerve block in animals [129,130] and humans ([131,132,133,134,135,136]; see [137] for a review). This is possibly mediated by a contraction of local vessels by the agonists, resulting in a decrease in the clearance of the anesthetics from the subarachnoid space [138,139]. Moreover, α_2_ agonists suppress nerve AP conduction and therefore exhibit a local anesthetic effect, contributing to enhanced local anesthetic effect [140]. For example, clonidine not only inhibits excitatory transmission in rat spinal lamina II neurons [141,142] but also blocks AP conduction in peripheral nerves [140,143,144]. The latter action requires a much higher concentration of clonidine than the former one. DEX as well as clonidine is reported to depress excitatory transmission in rat lamina II neurons [145]. Intracutaneous administration of DEX or clonidine together with lidocaine into the back of guinea-pigs increases in extent the local anesthetic effect of lidocaine [146]. Local wound infiltration with DEX added to bupivacaine more effectively alleviated postoperative pain compared to bupivacaine alone in humans [147]. It is possible that DEX as well as clonidine has a suppressive action on AP conduction, because DEX reportedly depresses voltage-gated Na^+^-channel currents [148].

DEX reduced the peak amplitude of frog sciatic nerve CAPs in a concentration-dependent manner in a range of 0.01–1 mM with an IC_50_ value of 0.40 mM [22]. Although DEX exhibited a high affinity for α_2_ adrenoceptors [123], DEX activity was not inhibited by α_2_-adrenoceptor antagonists, yohimbine and atipamezole ([128,149,150,151]; see [152,153] for reviews), indicating no involvement of α_2_ adrenoceptors [22]. CAP inhibition was also seen by other α_2_-adrenoceptor agonists, oxymetazoline (more selective to α_2A_ than α_2B_ and α_2C_; see [152,154] for reviews) and clonidine in a manner resistant to yohimbine. Oxymetazoline reduced CAP peak amplitude with an IC_50_ value of 1.5 mM; CAP amplitude was reduced by about 20% by clonidine at 2 mM [22]. This clonidine activity was different in extent from that (80% CAP amplitude reduction at 0.3 mM) reported previously for frog sciatic nerve CAPs [144], albeit a reason for this discrepancy is unknown. On the other hand, various adrenoceptor agonists, adrenaline, NA, α_1_-adrenoceptor agonist phenylephrine and β-adrenoceptor agonist isoproterenol at 1 mM had no effect on frog sciatic nerve CAPs [22]. A similar CAP inhibition produced by clonidine has been reported in the rat sciatic nerve. It has been demonstrated that CAPs originating from primary-afferent Aα and C fibers in the rat sciatic nerve are inhibited by clonidine with IC_50_ values of 2.0 and 0.45 mM, respectively [143].

The CAP inhibitory action of α_2_-adrenoceptor agonists would be due to an inhibition of voltage-gated Na^+^ and K^+^ channels involved in AP production. It has been reported that DEX inhibits voltage-gated Na^+^ channels in rat DRG neurons in a manner insensitive to yohimbine, although this type of Na^+^ channels is resistant to TTX [148]. IC_50_ value (0.058 mM) for this DEX activity in rat DRG neurons was about 10-fold smaller than that (0.40 mM) of frog CAP inhibition. The rat TTX-resistant Na^+^ channel was also inhibited by clonidine with an IC_50_ value of 0.26 mM [148]. TTX-sensitive Na^+^ channels in DRG neuroblastoma hybridoma cell line ND7/23 cells were inhibited by clonidine (IC_50_ = 0.824 mM; [155]). It has been reported in NG108-15 neuronal cells that delayed-rectifier K^+^-channels are inhibited by DEX with an IC_50_ value of 0.0046 mM and that TTX-sensitive Na^+^-channel current amplitudes are reduced by about 20% by DEX (0.01 mM) in a manner insensitive to yohimbine [156]. These distinctions in potency of drugs may be due to a difference in either animal species or Na^+^-channel types. Table 3 summarizes IC_50_ values for frog sciatic nerve fast-conducting CAP inhibitions produced by adrenoceptor agonists together with those for rat sciatic nerve CAPs and voltage-gated Na^+^ channels.

In clinical practice, DEX produces analgesia/sedation and decrease in heart rate, cardiac output and memory, each of whose actions depends on the plasma concentration of DEX in a distinct manner [157]. In patients, sedation is rapidly produced by 0.2 to 0.7 mg·kg^−1^·h^−1^ i.v. [123]; in intramuscular administration in cats, 40 mg·kg^−1^ is a usual dose for analgesia/sedation [158]. DEX concentrations, which are enough to suppress nerve AP conduction, are > 1000-fold higher than the use of DEX as α_2_ agonist, because the clinical use of DEX is <0.05 μM for plasma levels (see [157]). Therefore, the potential of DEX for nerve AP conduction block is independent of the use of DEX for analgesia/sedation. α_2_-Adrenoceptor agonists such as DEX, combined with a local anesthetic, have been used to extend peripheral nerve AP conduction block duration [130,132,133,134,135,159]. This effect is possibly due to a local vasoconstriction resulting in a delay of the absorption of the local anesthetic and/or a direct nerve AP conduction inhibition by α_2_ agonists [136]. The latter mechanism would be the above-mentioned nerve CAP inhibition produced by α_2_ agonists. This action makes sense when considering their topical application on nerves, but is not related to their use for analgesia/anesthesia by systemic administration. A chemical structure related to their α_2_ agonists (see [22]) may play a pivotal role in producing nerve AP conduction block.

### 3.2. Antiepileptics

Antiepileptics have various actions including GABA_A_-receptor activation, and voltage-gated Na^+^-, Ca^2+^-channel and glutamate-receptor inhibition (see [11,160] for reviews). As indicated by the inhibitory action on Na^+^ channel, nerve AP conduction inhibition is important for antiepileptics to alleviate neuropathic pain.

Frog sciatic nerve CAPs were inhibited by a phenyltriazine derivative (3,5-diamino- 6-(2,3-dichlorophenyl)-1,2,4-triazine; lamotrigine) that suppressed voltage-gated Na^+^ channels [161], and relieved central post-stroke pain and painful diabetic polyneuropathy [5]. This lamotrigine activity was partially reversible and concentration-dependent in a range of 0.02–0.5 mM with an IC_50_ value of 0.44 mM [23]. A similar CAP inhibition was seen by carbamazepine (5*H*-dibenz[b,f]azepine-5-carboxamide, an iminostilbene derivative [23]), which is different in chemical structure from lamotrigine while depressing voltage-gated Na^+^ channels [162]. Carbamazepine is known to be effective to relieve trigeminal neuralgia (see [163,164] for reviews). As different from lamotrigine, carbamazepine reduced frog sciatic nerve CAP amplitudes in a complete and reversible manner. This activity was concentration-dependent in a range of 0.05-1 mM with an IC_50_ value of 0.50 mM [23]. 10,11-Dihydro-10-oxo-5*H*-dibenz[b,f]-azepine- 5-carboxamide (oxcarbazepine, [165]), where there is a keto substitution at the 10,11 position of the dibenzazepine nucleus of carbamazepine, inhibited frog sciatic nerve CAPs with an efficacy smaller than that of carbamazepine [23]. Oxcarbazepine is reported to be effective in relieving painful diabetic neuropathy [5] and trigeminal neuralgia [163]. Oxcarbazepine activity was partially reversible and concentration-dependent in a range of 0.02–0.7 mM. When compared at 0.7 mM, CAP amplitude reduction by oxcarbazepine (40%) was somewhat smaller than that of carbamazepine (57%). Each of lamotrigine, carbamazepine and oxcarbazepine at 0.5 mM increased a threshold to evoke frog sciatic nerve CAPs [23]. This observation may be consistent with the fact that the antiepileptics reduce Na^+^-channel current amplitudes with a shift of their steady-state inactivation to a more negative membrane potential [166,167,168]. Another antiepileptic phenytoin (hydantoin derivative, 5,5-diphenylhydantoin; which inhibits voltage-gated Na^+^ channels [169] and alleviates paroxysm in trigeminal neuralgia [164]) reduced by a small extent frog sciatic nerve CAP peak amplitudes in a concentration-dependent manner in a range of 0.01–0.1 mM; this extent was only 15% at 0.1 mM [23].

When CAP and Na^+^ channel inhibition produced by antiepileptics were compared, the lamotrigine’s IC_50_ value (0.44 mM) in the frog sciatic nerve was close to that (0.641 mM at −90 mV) in inhibiting TTX-sensitive human brain type IIA Na^+^ channels expressed in Chinese hamster ovary cells [161]. Consistent with the fact that lamotrigine and carbamazepine had comparable IC_50_ values in the frog sciatic nerve, they reduced Na^+^-channel current amplitudes in N4TG1 mouse neuroblastoma cells with IC_50_ values similar to each other [168]. Oxcarbazepine at 0.5 mM reduced frog sciatic nerve CAP peak amplitudes by 20% [23]; this activity was much smaller than that for TTX-sensitive Na^+^-channel current reduction in differentiated NG108-15 neuronal cells (IC_50_ = 3.1 μM; [166]). The smaller CAP inhibition by oxcarbazepine than carbamazepine in the frog sciatic nerve may be consistent with the fact that oxcarbazepine was less effective than carbamazepine in suppressing seizures produced by maximal electroshock in rats [165]. The activity of phenytoin (CAP amplitude reduction of 15% at 0.1 mM) in the frog sciatic nerve was less than those for rat cortical and human type IIA Na^+^ channels (60–90% amplitude reduction at −60 mV by phenytoin at 0.1 mM) [161,169]. As different from frog sciatic nerve CAPs, phenytoin reduced Na^+^ channel current amplitudes in N4TG1 mouse neuroblastoma cells with an IC_50_ value similar to that of lamotrigine [168]; phenytoin, lamotrigine and carbamazepine were reported to bind to a common site of Na^+^ channels in rat hippocampal CA1 neurons [167]. A sensitivity of voltage-gated Na^+^ channels to phenytoin may be different in extent among distinct types of the channel. Consistent with this idea, phenytoin actions differed among human Nav1.1, Nav1.2, Nav1.3 and Nav1.4 α-subunits (all of which are TTX-sensitive) expressed in HEK293 cells [170]; there was a distinction in the properties and accessibilities of Na^+^ channels between frog and rat myelinated nerves [171].

Antiepileptics having an ability to suppress CAPs are similar in chemical structure to NSAIDs in that lamotrigine, carbamazepine and oxcarbazepine have two unsaturated six-membered rings (see Figures 1a, 2aA and 2bA in [23] for the chemical structures of the three antiepileptics). Carbamazepine and diclofenac appear to have a common or closely related binding site owing to an occlusion of their effects on voltage-gated Na^+^ channels [113].

On the other hand, frog sciatic nerve CAPs were not affected by other antiepileptics, gabapentin (1-(aminomethyl)cyclohexaneacetic acid; which is related to GABA in chemical structure and relieves post-herpetic neuralgia [164]), topiramate (2,3:4,5-bis-*O*-(1-methylethylidene)- β-D-fructopyranose sulfamate; which alleviates various neuropathic pains including intercostal neuralgia and trigeminal neuralgia [11]) and sodium valproate (2-propylpentanoic acid sodium salt; which relieves diabetic neuropathic pain [11]), at a high concentration such as 10 mM [23].

The less effectiveness of gabapentin and sodium valproate in the frog sciatic nerve was similar to that for human type IIA Na^+^ channels [161]. Gabapentin at concentrations of <3 mM hardly affected the human Na^+^ channels [161]. Gabapentin’s antinociceptive action would be mainly due to its binding to the α_2_δ subunit of voltage-gated Ca^2+^ channels, leading to an inhibition of Ca^2+^ entry in nerve terminals which in turn suppresses the release of neurotransmitters from there (see [172] for review). As different from the frog sciatic nerve, topiramate reduced TTX-sensitive Na^+^-channel current amplitudes with an IC_50_ value of 0.0489 mM in rat cerebellar granule cells [173]. Such a distinction would be possibly attributed to a difference in topiramate sensitivity among different types or phosphorylation states of Na^+^ channels [174]. Antinociceptive actions of sodium valproate and topiramate have been attributed to other mechanisms such as GABA_A_-receptor response increase (see [175,176] for reviews). Glutamate-receptor inhibition also would possibly contribute to the antinociceptions produced by topiramate and lamotrigine, because topiramate suppresses GluK1 (GluR5) kainate receptors (a subtype of glutamate receptors) in rat basolateral amygdala neurons [177] and lamotrigine depresses α-amino-3-hydroxy-5-methyl-4-isoxazole propionate (AMPA) receptors (another subtype of glutamate receptors) in rat dentate gyrus granule cells [178]. Table 4 summarizes IC_50_ values for frog sciatic nerve fast-conducting CAP inhibitions produced by antiepileptics together with those for voltage-gated Na^+^ channels.

Antiepileptics having an ability to suppress frog sciatic nerve CAPs appeared to have antinociceptive actions in a persistent pain model. Intraperitoneal administration of lamotrigine, carbamazepine and oxcarbazepine produced analgesic effects in the second phase of the formalin test whereas phenytoin, topiramate and sodium valproate did not in rats [179,180]. The antinociceptive effects of antiepileptics seemed to be related to nerve AP conduction inhibition produced by them. The plasma concentrations of lamotrigine and carbamazepine used to clinically treat epilepsy are, respectively, <12 and 20–50 μM [181,182], values smaller than those of IC_50_ for frog sciatic nerve CAP inhibition.

### 3.3. Antidepressants

Cellular mechanisms for antinociception produced by antidepressants are thought to be activation of the 5-HT- and NA-containing descending antinociceptive pathway to spinal dorsal horn through an inhibition of the reuptake of their neurotransmitters [183,184], involvement of α adrenoceptors, H_1_-histamine, 5-HT, opioid and muscarinic acetylcholine receptors ([9,185,186,187,188]; see [189,190] for reviews) and inhibition of voltage-gated Ca^2+^ [191,192], *N*-methyl-D-aspartate (NMDA)-receptor (a subtype of glutamate receptors; [193,194,195]) and P2X_4_-receptor channels (a subtype of ionotropic P2X receptors; [196]), all of which are related to synaptic transmission.

Frog sciatic nerve CAPs were inhibited by a 5-HT and NA reuptake inhibitor (SNRI) duloxetine ([197,198,199]; see [200] for review) in a partially reversible manner. Duloxetine activity was concentration-dependent in a range of 0.001–2 mM with an IC_50_ value of 0.23 mM [24]. A similar CAP inhibition was produced by a selective 5-HT reuptake inhibitor (SSRI) fluoxetine ([185,186]; see [190,201] for reviews). Fluoxetine reduced frog sciatic nerve CAP peak amplitudes; this activity was partially reversible, concentration-dependent in a range of 0.05–5 mM and had an IC_50_ of 1.5 mM, a value larger than that of duloxetine [24].

Typical tricyclic antidepressants (amitriptyline and desipramine, which are tertiary and secondary amines, respectively; [184,187,202,203]) exhibited a similar inhibitory action on frog sciatic nerve CAPs. CAP peak amplitudes were reduced by amitriptyline in a concentration range of 0.001–1 mM with an IC_50_ value of 0.26 mM, and by desipramine in a concentration range of 0.1–5 mM with an IC_50_ value of 1.6 mM [24]. Thus, amitriptyline was six-fold more effective than desipramine in inhibiting CAPs. The activity of amitriptyline was consistent with its AP conduction blockade action in the rat sciatic nerve [204].

As with tricyclic antidepressants, a tetracyclic one maprotiline [202] also reduced frog sciatic nerve CAP peak amplitudes in a partially reversible manner. Maprotiline activity was concentration-dependent in a range of 0.2–5 mM with an IC_50_ value of 0.95 mM [24]. Trazodone, which is known as 5-HT 2 antagonist and reuptake inhibitor (SARI), is a non-SNRI, -SSRI, -tricyclic and -tetracyclic antidepressant ([205,206,207,208]; see [209] for review). Frog sciatic nerve CAPs were inhibited by trazodone at concentrations ranging from 0.2 to 2 mM, a maximally dissolved one, in a partially reversible manner. The extent of CAP peak amplitude reduction by trazodone at 1.0 mM was about 50% [24].

The CAP amplitude reductions produced by the antidepressants would be mediated by an inhibition of TTX-sensitive voltage-gated Na^+^ channels which are involved in producing frog sciatic nerve CAPs. This idea is supported by the observations that voltage-gated Na^+^ channels were suppressed by duloxetine [210,211], fluoxetine [212], amitriptyline [210,212,213,214,215,216,217,218], desipramine and maprotiline [219]. TTX-sensitive Na^+^ channels in bovine adrenal chromaffin cells were inhibited by amitriptyline (IC_50_ = 0.0202 mM), fluoxetine (62% amplitude reduction at 0.02 mM), desiparmine (50% at 0.02 mM) and trazodone (20% at 0.1 mM; [212]); amitriptyline also inhibited Na^+^ channels in rat clonal pituitary GH_3_ cells with an IC_50_ value of 0.0398 mM [204]. These efficacies in inhibiting Na^+^ channels were much larger than those of frog sciatic nerve CAPs. Furthermore, IC_50_ value (0.0221 mM) for duloxetine to reduce (TTX-sensitive) Nav1.7 channel current amplitudes was about 10-fold less than that (0.23 mM) for frog sciatic nerve CAP inhibition [211], but the efficacy sequence for CAP inhibition (maprotiline > fluoxetine) was the same as that of the Nav1.7 channel, where IC_50_ values for maprotiline, fluoxetine, desipramine and amitriptyline were 0.028, 0.074, 0.024 and 0.085 mM, respectively [219]. The observation that amitriptyline and duloxetine had a comparable IC_50_ value for frog sciatic nerve CAP inhibition was the same as that for cardiac-type Na^+^-channel inhibition [210]. TTX-resistant Na^+^ channels (possibly Nav1.8 channels) in rat trigeminal ganglion neurons were also inhibited by amitriptyrine with an IC_50_ value of 0.00682 mM [220]. Typical local anesthetics have hydrophobic and hydrophilic moieties that are separated by an intermediate ester or amide linkage (see [221] for review), while all of the antidepressants tested in the frog sciatic nerve, except for trazodone, have a hydrophilic amine group and a hydrophobic moiety containing benzene rings, both of which are linked by a straight chain hydrocarbon, in their chemical structures (see Figure 1 in [24] for the chemical structures of six antidepressants tested). Such chemical structures may take an important role in Na^+^ channel inhibition. Table 5 summarizes IC_50_ values for frog sciatic nerve fast-conducting CAP inhibitions produced by antidepressants together with those for voltage-gated Na^+^ channels.

The antidepressants tested are clinically used to treat chronic pain [8,9,198,200,203,222,223] and inhibit neuropathic pain in animal models. For instance, duloxetine suppressed tactile allodynia and heat hyperalgesia in neuropathic pain rat models [197]; fluoxetine produced antinociception in streptozotocin-induced diabetic neuropathic pain mouse models [185]; amitriptyline and desipramine were effective in alleviating pain in patients with diabetic neuropathy [184]; maprotiline depressed neuropathic pain produced by chronic constriction injury of the sciatic nerve in rats [224]; trazodone inhibited hyperalgesia in chronic constriction injury rat models [206]. The plasma concentrations of duloxetine, fluoxetine, amitriptyline, desipramine, maprotiline and trazodone used to clinically treat depression and neuropathic pain are, respectively, 0.09–0.3, 0.3–1.6, 0.36–0.90, 0.47–1.1, 0.72–1.4 and 2.2–4.3 μM [189,211], values much smaller than those of IC_50_ for frog sciatic nerve CAP inhibition.

### 3.4. Local Anesthetics

Local anesthetics have been used for the treatment of the neuropathic pain with an expectation of the inhibition of nerve AP conduction in animals [225,226] and humans [227,228,229,230], although other effects including neurotransmitter receptor and TRP channel activation are possibly involved in antinociception [231]. Many of local anesthetics reduce both voltage-gated Na^+^ and K^+^ channel current amplitudes ([71]; see [76,231,232] for reviews).

An amide-type local anesthetic lidocaine, which is known to block AP conduction [58,59,60,232], reversibly reduced the peak amplitude of frog sciatic nerve CAPs in a wide concentration range of 0.1 to 2 mM with an IC_50_ value of 0.74 mM [19]. This value was somewhat larger than that (0.204 mM) for Na^+^-channel current amplitude reduction in *Xenopus laevis* sciatic nerve fibers [71]. Voltage-gated K^+^ channels in this preparation were also inhibited by lidocaine with an IC_50_ value (1.118 mM) larger than that for Na^+^ channels [71]. Rat TTX-resistant Na^+^ channel was inhibited by lidocaine with an IC_50_ of 0.073 mM [148], a value 10-fold lower than that for frog sciatic nerve CAP inhibition.

A similar reversible CAP inhibition was produced by another amide-type local anesthetic ropivacaine, which has a longer duration of action in terms of nerve AP conduction block than lidocaine does ([233]; see [234] for review), in a concentration range of 0.01–1 mM with an IC_50_ value of 0.34 mM [19]. CAP peak amplitude reduction produced by ropivacaine in the frog sciatic nerve was almost similar in extent to that (about 30% at 0.2 mM) reported for rabbit vagus nerve A fibers [235]. IC_50_ values for lidocaine and ropivacaine (0.74 and 0.34 mM, respectively) in the frog sciatic nerve were not so different from those (0.28 mM for both lidocaine and ropivacaine) for fast-conducting CAPs in the rat sciatic nerve [236]. Moreover, an amide-type local anesthetic prilocaine also reversibly reduced frog sciatic nerve CAP peak amplitudes; this activity was concentration-dependent in a range of 0.01–5 mM with an IC_50_ value of 1.8 mM [26].

Levobupivacaine and its racemic bupivacaine are amide-type local anesthetics; the former has a lower risk of cardiovascular and CNS toxicity than the latter ([237]; see [238] for review). Levobupivacaine reversibly reduced frog sciatic nerve CAP peak amplitudes in a concentration range of 0.05–1 mM with an IC_50_ value of 0.23 mM [23]. This value was almost comparable to IC_50_ value (0.22 mM) reported previously for tonic inhibition of frog sciatic nerve CAPs by levobupivacaine [237] and to that (0.264 mM) for tonic inhibition by this drug of voltage-gated Na^+^-channel currents recorded at −100 mV in GH-3 neuroendocrine cells [239]. As reported previously [237], the levobupivacaine activity in the frog sciatic nerve was smaller than that of bupivacaine (CAP amplitude reductions at 0.5 mM: 45 and 76%, respectively; [23]). This bupivacaine activity was smaller than that (IC_50_ = 0.027 mM) for Na^+^-channel current amplitude reduction in *Xenopus laevis* sciatic nerve fibers [71], that (IC_50_ = 0.178 mM) for TTX-sensitive Na^+^ channels in DRG neuroblastoma hybridoma cell line ND7/23 cells [155] and that (IC_50_ = 0.190 mM) for Na^+^ channels in rat clonal pituitary GH_3_ cells [204]. Voltage-gated K^+^ channels in *Xenopus laevis* sciatic nerve fibers were also inhibited by bupivacaine with an IC_50_ value (0.092 mM) larger than that for Na^+^ channels [71].

Cocaine, a compound derived from the coca plant *Erythroxylon coca*, is a classic ester-type local anesthetic which is well-known to inhibit nerve AP conduction ([68,240,241]; see [242] for review). Cocaine reversibly reduced frog sciatic nerve CAP peak amplitudes in a reversible manner. This cocaine activity was concentration-dependent in a range of 0.01–2 mM with an IC_50_ of 0.80 mM [20], a value almost comparable to that (0.74 mM) of lidocaine in the frog sciatic nerve [19]. The IC_50_ value for cocaine was 4-fold larger than that (about 0.2 mM) in the rat phrenic nerve [68]. Although mouse phrenic CAP peak amplitudes were reduced by 26% by cocaine (40 μM [241]), such a reduction in the frog sciatic nerve was produced at a concentration of about 300 μM [20]. When compared between the frog and rat sciatic nerve, there was an almost similar CAP amplitude reduction by cocaine (frog: 30% at 0.5 mM; rat: 40% at 0.375 mM; see [72]). Much evidence shows an inhibition of voltage-gated Na^+^ channels by cocaine (for example, see [72,243,244]), where a tonic (TTX-resistant) Nav1.5 channel current amplitude reduction produced by cocaine (0.05 mM) is ca. 70% [244]. Cocaine and lidocaine exhibit a competitive interaction at Na^+^ channels [245]. Cocaine is reported to also depress delayed rectifier K^+^ channels in central snail neurons [246].

A well-known another ester-type local anesthetic procaine [247] reversibly reduced frog sciatic nerve CAP peak amplitudes in a concentration range of 0.1–5 mM with an IC_50_ value of 2.2 mM [27]. This value was almost comparable to those (2–5 mM) obtained by other investigators [248,249] in the same preparation and also to that (ca. 1 mM) reported previously in the rat sciatic nerve [248]. Moreover, a ratio of the procaine’s IC_50_ value to that of lidocaine (0.74 mM; [19]) for frog CAP inhibition was comparable to a ratio of procaine concentration (0.53%), needed for 50% motor nerve conduction block, to lidocaine’s one (0.14%) in rats [250]. On the other hand, the frog sciatic nerve procaine activity was 37-fold smaller than that (IC_50_ = 0.060 mM) for Na^+^-channel current amplitude reduction in *Xenopus laevis* sciatic nerve fibers [71]. Voltage-gated K^+^ channels in this preparation were also inhibited by procaine with an IC_50_ value (6.303 mM) larger than that for Na^+^ channels [71].

An ester-type local anesthetic benzocaine (ethyl 4-aminobenzoate) is used for topical anesthesia in clinical medicine (see [251] for review) and also for amphibian anesthesia ([252]; see [253,254] for reviews). Benzocaine reversibly reduced frog sciatic nerve CAP peak amplitudes; this activity was concentration-dependent in a range of 0.01–2 mM with an IC_50_ value of 0.80 mM (73% reduction at 1 mM; [25]). This activity was similar to that for CAP inhibition in the rat sciatic nerve (37% inhibition at 1.3 mM; [59]). When compared with other local anesthetics, benzocaine activity was similar to those of cocaine and lidocaine.

Frog CAP peak amplitude was reduced by an ester-type local anesthetic tetracaine in a reversible and concentration-dependent manner with an IC_50_ value of 0.014 mM. This value is not so distinct from that (0.0063 mM) of frog sciatic nerve fibers, as reported previously [144], and also from that (0.009 mM) of rabbit A nerve fibers [255]. On the other hand, the frog sciatic nerve tetracaine activity was 19-fold smaller than that (IC_50_ = 0.0007 mM) for Na^+^-channel current reduction in *Xenopus laevis* sciatic nerve fibers [71]. Voltage-gated K^+^ channels in this preparation were also inhibited by tetracaine with an IC_50_ value (0.946 mM) much larger than that for Na^+^ channels [71]. Tetracaine had much more effectiveness than lidocaine, bupivacaine and procaine in both frog CAP and *Xenopus laevis* Na^+^-channel current inhibition.

A non-amide- and non-ester-type local anesthetic pramoxine also reduced frog sciatic nerve CAP peak amplitudes with a slow recovery to those before its application. This pramoxine activity was concentration-dependent in a range of 0.001–1 mM with an IC_50_ value of 0.21 mM [26]. Table 6 summarizes IC_50_ values for frog sciatic nerve fast-conducting CAP inhibitions produced by local anesthetics together with those for rat sciatic nerve CAPs and voltage-gated Na^+^ channels. The chemical structures of local anesthetics tested are shown in Figure 4 in [25].

## 4. Comparison in the Efficacy of Nerve Conduction Inhibition among Analgesics and Analgesic Adjuvants

Some of analgesic adjuvants inhibited frog sciatic nerve CAPs with similar IC_50_ values. For instance, antidepressants’ IC_50_ values were similar to those of some of antiepileptics, α_2_-adrenoceptor agonists and local anesthetics. Duloxetine and amitriptyline values (0.23 and 0.26 mM, respectively; see Table 5) were close to those of lamotrigine, carbamazepine, DEX, ropivacaine, levobupivacaine and pramoxine (0.44, 0.50, 0.40, 0.34, 0.23 and 0.21 mM, respectively; Table 3, Table 4 and Table 6). On the other hand, fluoxetine, desipramine, maprotiline and trazodone values (1.5, 1.6, 0.95 and ca. 1 mM, respectively; Table 5) were similar to those of oxymetazoline, lidocaine, cocaine, procaine and prilocaine (1.5, 0.74, 0.80, 2.2 and 1.8 mM, respectively; Table 3 and Table 6). In each of the former (IC_50_: 0.2–0.5 mM) and latter (IC_50_: 1–2 mM) groups, a chemical structure common among many drugs was not noted, although there was a relationship between CAP inhibition extent and the number of CH_2_ in opioids having similar structures, as mentioned in Section 2.1. The IC_50_ values of the antidepressants were much larger than that of tetracaine (0.014 mM; Table 6). Thus, some of the analgesic adjuvants will have an ability to inhibit nerve conduction with an efficacy comparable to each other.

When antipyretic analgesics NSAIDs were compared with analgesic adjuvants, diclofenac’s IC_50_ value (0.94 mM; Table 2) was similar to those of maprotiline, trazodon, lidocaine and cocaine (0.95, ca. 1, 0.74 and 0.80 mM, respectively; Table 5 and Table 6), while aceclofenac, tolfenamic acid, meclofenamic acid and flufenamic acid (0.47, 0.29, 0.19 and 0.22 mM, respectively; Table 2) had IC_50_ values being close to those of duloxetine, amitriptyline, lamotrigine, carbamazepine, DEX, ropivacaine, levobupivacaine and pramoxine (0.23, 0.26, 0.44, 0.50, 0.40, 0.34, 0.23 and 0.21 mM, respectively; Table 3, Table 4, Table 5 and Table 6). The NSAIDs’ values were smaller than those of fluoxetine, desipramine, oxymetazoline, procaine and prilocaine (1.5, 1.6, 1.5, 2.2 and 1.8 mM, respectively; Table 3, Table 5 and Table 6) while being larger than that of tetracaine (0.014 mM; Table 6). Thus, NSAIDs could inhibit nerve AP conduction with efficacies comparable to some of analgesic adjuvants. In these cases also, no common chemical structure was found among compounds with similar IC_50_ values.

Not only analgesic adjuvants and NSAIDs but also narcotic analgesics opioids have an ability to depress nerve AP conduction. Opioids reduced frog sciatic nerve CAP peak amplitudes; morphine and codeine at 5 mM reduced CAP peak amplitude by 15% and 30%, respectively, and tramadol and ethylmorphine had the IC_50_ values of 2.3 and 4.6 mM, respectively (Table 1). These opioid actions were smaller in magnitude than those of analgesic adjuvants and NASIDs. For example, the tramadol’s IC_50_ value (2.3 mM) was larger by 3.1- and 6.8-fold than those (0.74 mM and 0.34 mM, respectively) of lidocaine and ropivacaine, respectively [19]. It has been previously reported that lidocaine reduces frog sciatic nerve CAP amplitudes with an IC_50_ of 6.6 mM [60], a value larger by three-fold than the tramadol value. However, ratio of the IC_50_ value of tramadol to that of lidocaine was almost comparable to Katsuki et al. [19]’s one, albeit IC_50_ values for lidocaine were largely distinct between the two studies [19,60]. A contribution of nerve AP conduction suppression to analgesia produced by opioids appears to be much less in extent compared to other well-known cellular mechanisms such as a decrease in the release of L-glutamate from nerve terminals and a membrane hyperpolarization in the spinal dorsal horn (for example, see [28,29]). Thus, nerve AP conduction inhibition may be a common mechanism for antinociception induced by analgesic adjuvants and NSAIDs but not opioids.

IC_50_ values for analgesic adjuvants in frog sciatic nerve CAPs were comparable in local anesthetic sensitivity to those in rat sciatic nerve CAPs while being generally larger than IC_50_ values for TTX-sensitive Na^+^ channel currents. There are several possibilities for this difference. First, not only voltage-gated Na^+^ but also K^+^ channels are involved in the production of the CAP. Second, there may be a difference in expressed TTX-sensitive Na^+^-channel types (Nav1.1–1.4, Nav1.6 and Nav1.7) among the preparations examined. Third, CAPs are measured from the bundle of nerve fibers while Na^+^ currents from single cells. Since the analgesic adjuvants used clinically act on the nerve trunk and nerve conduction is mediated by both voltage-gated Na^+^ and K^+^ channels, their sciatic nerve IC_50_ values may be an appropriate measure for a nerve conduction inhibition *in vivo*. Considering that nociceptor-specific deletion of TTX-sensitive Nav1.7 gene results in attenuated acute and inflammatory hyperalgesia in mice [256], Na^+^ channels may be the main target of analgesic adjuvants.

## 5. Antinociceptive Plant-Derived Compounds Inhibit Nerve Conduction with Efficacies Comparable with Those of Analgesic Adjuvants and NSAIDs

CAP inhibitory actions similar to those of analgesic adjuvants and NSAIDs were seen by plant-derived compounds that are known to produce antinociception by their oral, intraperitoneal or intrathecal administration (see [257,258] for reviews). Thus, frog sciatic nerve CAPs were inhibited by plant-derived compounds whose IC_50_ values were close to those of analgesic adjuvants and NSAIDs ([27,259,260,261]; see [262] for review). Carvacrol, thymol, citronellol, bornyl acetate, citral, citronellal and geranyl acetate had IC_50_ values of 0.34, 0.34, 0.35, 0.44, 0.46, 0.50 and 0.51 mM, respectively, in reducing CAP peak amplitudes. These values were similar to those of duloxetine (0.23 mM), amitriptyline (0.26 mM), aceclofenac (0.47 mM), tolfenamic acid (0.29 mM), meclofenamic acid (0.19 mM) and flufenamic acid (0.22 mM; see Table 2 and Table 5). On the other hand, IC_50_ values of (+)-pulegone, (−)-carvone, (+)-borneol, (−)-menthone, cinnamaldehyde and allyl isothiocyanate (1.4, 1.4, 1.5, 1.5, 1.2 and 1.5 mM, respectively) were comparable to those of fluoxetine (1.5 mM) and desipramine (1.6 mM; Table 5). IC_50_ values for diclofenac, maprotiline and trazodone (0.94, 0.95 and ca. 1.0 mM, respectively; Table 2 and Table 5) were close to those of linalyl acetate, eugenol and (−)-menthol (0.71, 0.81 and 1.1 mM, respectively). Such CAP inhibitions produced by plant-derived compounds are possibly due to a suppression of TTX-sensitive voltage-gated Na^+^ channels (for example, [263,264,265,266]; see [267] for review). Altogether, plant-derived chemicals could replace analgesic adjuvants and NSAIDs in terms of nerve AP conduction inhibition.

Frog sciatic nerve CAPs were inhibited by a seven-membered ring compound hinokitiol (β-thujaplicin; 2-hydroxy-4-isopropylcyclohepta-2,4,6-trien-1-one) contained in a species of cypress tree [268] with an IC_50_ of 0.54 mM, a value comparable to those of many plant-derived compounds. This inhibition was possibly due to an interaction involving its carbonyl, isopropyl and hydroxyl groups [269]. This carbonyl bond of hinokitiol serves for its seven-membered ring to act as a benzene ring, while the isopropyl and hydroxyl groups are important for the hinokitiol-induced CAP inhibition. Consistent with this idea, benzene-ring compounds having the isopropyl and hydroxyl groups, such as thymol, carvacrol, biosol (a stereoisomer of thymol and carvacrol; IC_50_ = 0.58 mM) and 4-isopropylphenol (0.85 mM), had an ability to inhibit frog sciatic nerve CAPs (see above; [259,269]). Hinokitiol exhibits various actions such as inhibition of apoptosis [270], anti-bacterial, anti-inflammatory [271], insecticidal [272], anti-fungal [273], anti-tumor [274] and cytotoxic activities [275,276]. Hinokitiol used as a dermatological drug to suppress inflammation may exhibit a local anesthetic effect. Administration of an oral care gel containing hinokitiol to the oral mucosa reportedly alleviated oral pain in patients with oral lichen planus associated with hepatitis C virus infection [277]. Such a pain alleviation may be partly due to a local anesthetic effect of hinokitiol.

Moreover, a general anesthetic propofol (2,6-diisopropylphenol; [278,279,280]; see [281,282] for reviews) having two isopropyl groups and one hydroxyl group bound to the benzene ring was found to concentration-dependently inhibit frog sciatic nerve CAPs with an IC_50_ value of 0.14 mM [25]. Consistent with this result, propofol is reported to inhibit APs recorded extracellularly in the human and mammalian CNS [283,284].

Traditional Japanese (Kampo) medicines that are composed of plant-derived crude drugs are used together with Western medicines in Japan with various purposes including antinociception (see [285,286,287,288,289] for reviews). Frog sciatic nerve CAPs were concentration-dependently inhibited by Kampo medicines, daikenchuto, rikkosan, kikyoto, rikkunshito, shakuyakukanzoto and kakkonto; among them, daikenchuto was the most effective with an IC_50_ value of 1.1 mg/mL. With respect to the activities of three kinds of crude medicine contained in daikenchuto, CAPs were inhibited by Japanese pepper and processed ginger while being hardly affected by ginseng radix. Japanese pepper’s IC_50_ value was 0.77 mg/mL and the extent of CAP peak amplitude reduction produced by processed ginger at 2 mg/mL was 31% [290]. At least a part of Kampo medicine’s antinociceptive effect could attribute to its nerve AP conduction inhibitory action.

## 6. Conclusions

This review article demonstrated that some of NSAIDs, analgesic adjuvants and plant-derived compounds having analgesic activities inhibit frog sciatic nerve CAPs with similar efficacies. Although the CAPs are fast-conducting TTX-sensitive Aα fiber-mediated ones, nociceptive information is transmitted by slow-conducting Aδ and C fibers [1]. In the frog sciatic nerve, Aδ-fiber CAPs were not able to be isolated from Aα-fiber ones; C-fiber CAPs were much smaller in peak amplitude and conduction velocity than fast-conducting ones [18] and thus were not able to be recorded. Therefore, the effects of the antinociceptive drugs on slow-conducting CAPs were not examined. In order to more elucidate a difference in the extent of nerve AP conduction inhibition among various antinociceptive drugs, it would be necessary to investigate their effects on slow-conducting CAPs.

In preparations other than the frog sciatic nerve, antinociceptive drugs are reported to inhibit not only A-fiber but also C-fiber CAPs. For example, in the rabbit vagus nerve, fentanyl and sufentanil reduced C-fiber CAP amplitudes with extents smaller than those of A-fiber ones [39] and lidocaine blocked nerve conduction in not only myelinated A- but also unmyelinated C-fibers ([291]; A-fiber CAPs were more sensitive to lidocaine than C-fiber’s ones in rats, [292]). Clonidine inhibited both Aα-fiber and C-fiber CAPs (see Section 3.1). Moreover, many investigators have reported an inhibition by lidocaine, α_2_-adrenoceptor agonists [148] and NSAIDs [108,115,293] of TTX-resistant voltage-gated Na^+^ channels (see Table 2, Table 3 and Table 5) that may be involved in producing slow-conducting CAPs. It has been reported that the knockdown of TTX-resistant Nav1.8 channels results in inhibition of neuropathic and inflammatory pain in rats [294].

Since the concentrations of analgesics and analgesic adjuvants necessary to inhibit CAPs are higher than their clinically relevant ones, as mentioned in the previous sections, nerve conduction inhibition may occur when the drugs is administrated locally or is accumulated in the nervous system. If such an inhibition occurs in Aα fibers innervating skeletal muscle, this would result in undesirable side effects such as muscle paralysis. C and Aδ fibers are smaller in diameter than Aα ones; therefore, if the drugs act on voltage-gated Na^+^ channels from cytoplasm side, C fibers will precede Aα fibers in inhibiting nerve conduction owing to a difference in surface-to-volume ratio between the fibers. Thus, the drugs will have to be used at the lowest possible concentrations. It is suggested that at least a part of antinociception produced by analgesics and analgesic adjuvants is attributed to their inhibitory actions on nerve AP conduction mediated by voltage-gated Na^+^ channels that are sensitive and resistant to TTX.

## Figures and Tables

**Table 1 pharmaceuticals-13-00062-t001:** Comparison of IC_50_ values in inhibiting frog or rat sciatic nerve fast-conducting CAPs and TTX-sensitive Na^+^ channels among opioids.

Opioids	Frog CAP	Rat CAP	TTX-Sensitive	References
	IC_50_ (mM)	IC_50_ (mM)	Na^+^ Channel Current	
			IC_50_ (mM)	
Tramadol	2.3	37% reduction	0.194	[19,59,64]
		(4 mM)	0.103	[65]
Mono-*O*-desmethyl -tramadol	9% reduction			[19]
	(5 mM)			
Morphine	15% reduction		0.378	[20,64]
	(5 mM)			
Codeine	30% reduction			[20]
	(5 mM)			
Ethylmorphine	4.6			[20]

Here, where IC_50_ values are not available, it is partly shown for comparison how CAP amplitudes are reduced by drugs, where % value indicates the extent of the reduction at the concentration shown in parentheses.

**Table 2 pharmaceuticals-13-00062-t002:** Comparison of IC_50_ values in inhibiting frog sciatic nerve fast-conducting CAPs, TTX-sensitive or -resistant Na^+^ channels among NSAIDs.

NSAIDs		Frog CAP IC_50_ (mM)	TTX-Sensitive Na^+^ Channel Current IC_50_ (mM)	TTX-Resistant Na^+^ Channel Current IC_50_ (mM)	References
Acetic Acid-Based	Diclofenac	0.94	0.00851, 0.014	ca. 20% reduction (0.3 mM)	[21,108,110,115]
	Aceclofenac	0.47			[21]
	Indomethacin	38% reduction (1 mM)			[21]
	Acemetacin	38% reduction (0.5 mM)			[21]
	Etodolac	15% reduction (1 mM)			[21]
	Sulindac	n.d. (no effect, 1 mM)			[21]
	Felbinac	n.d. (no effect, 1 mM)			[21]
Fenamic Acid-Based	Tolfenamic acid	0.29	ca. 70% reduction (0.1 mM)	ca. 30% reduction (0.1 mM)	[21,116]
	Meclofenamic acid	0.19			[21]
	Mefenamic acid	16% reduction (0.2 mM)			[21]
	Flufenamic acid	0.22	ca. 60% reduction (0.1 mM)	ca. 30% reduction (0.1 mM)	[21,116]
			0.189		[114]
Salicylic Acid-Based	Aspirin	n.d. (no effect, 1 mM)			[21]
Propionic Acid-Based	Ketoprofen	n.d. (no effect, 1 mM)			[21]
	Naproxen	n.d. (no effect, 1 mM)			[21]
	Ibuprofen	n.d. (no effect, 1 mM)			[21]
	Loxoprofen	n.d. (no effect, 1 mM)			[21]
	Flurbiprofen	n.d. (no effect, 1 mM)			[21]
Enolic Acid-Based	Meloxicam	n.d. (no effect, 0.5 mM)			[21]
	Piroxicam	n.d. (no effect, 1 mM)			[21]

Here, when IC_50_ values are not available, it is partly shown for comparison how the CAPs and channels are affected by drugs, where % value indicates the extent of the reduction at the concentration shown in parentheses; n.d.: not determined.

**Table 3 pharmaceuticals-13-00062-t003:** Comparison of IC_50_ values in inhibiting frog or rat sciatic nerve fast-conducting CAPs and TTX-sensitive or -resistant Na^+^ channels among adrenoceptor agonists.

Adrenoceptor Agonists		Frog CAP IC_50_ (mM)	Rat CAP IC_50_ (mM)	TTX-Sensit. Na^+^ Channel Current IC_50_ (mM)	TTX-Resist. Na^+^ Channel Current IC_50_ (mM)	References
	Adrenaline	n.d. (no effect, 1 mM)				[22]
	Noradrenaline	n.d. (no effect, 1 mM)				[22]
α_2_ Agonist	Dexmedetomidine	0.40		ca. 20% reduct. (0.01 mM)	0.058	[22,148,156]
	Oxymetazoline	1.5				[22]
	Clonidine	ca. 20% reduct. (2 mM)	2.0	0.824	0.26	[22,143,148,155]
α_1_ Agonist	Phenylephrine	n.d. (no effect, 1 mM)				[22]
β Agonist	Isoproterenol	n.d. (no effect, 1 mM)				[22]

Here, when IC_50_ values are not available, it is partly shown for comparison how the CAPs and channels are affected by drugs. sensit.: sensitive; resist.: resistant; reduct.: reduction (where % value indicates the extent of the reduction at the concentration shown in parentheses); n.d.: not determined.

**Table 4 pharmaceuticals-13-00062-t004:** Comparison of IC_50_ values in inhibiting frog sciatic nerve fast-conducting CAPs and TTX-sensitive Na^+^ channels among antiepileptics.

Antiepileptics	Frog CAP IC_50_ (mM)	TTX-Sensitive Na^+^ Channel Current IC_50_ (mM)	References
Lamotrigine	0.44	0.641 (at −90 mV)	[23,161]
Carbamazepine	0.50		[23]
Oxcarbazepine	20% reduction (0.5 mM)	0.0031	[23,166]
Phenytoin	15% reduction (0.1 mM)	60–90% reduction (0.1 mM at −60 mV)	[23,161,169]
Gabapentin	n.d. (no effect, 10 mM)	no effect (<3 mM)	[23,161]
Topiramate	n.d. (no effect, 10 mM)	0.0489	[23,173]
Sodium valproate	n.d. (no effect, 10 mM)		[23]

Here, when IC_50_ values are not available, it is partly shown for comparison how the CAPs and channels are affected by drugs, where % value indicates the extent of the reduction at the concentration shown in parentheses; n.d.: not determined.

**Table 5 pharmaceuticals-13-00062-t005:** Comparison of IC_50_ values in inhibiting frog sciatic nerve fast-conducting CAPs and TTX-sensitive or -resistant Na^+^ channels among antidepressants.

Antidepressants	Frog CAP IC_50_ (mM)	TTX-Sensitive Na^+^ Channel Current IC_50_ (mM)	TTX-Resistant Na^+^ Channel Current IC_50_ (mM)	References
Duloxetine	0.23	0.0221		[24,211]
Fluoxetine	1.5	0.074		[24,219]
		62% reduction (0.02 mM)		[212]
Amitriptyline	0.26	0.0202	0.00682	[24,212,220]
		0.0398		[204]
		0.085		[219]
Desipramine	1.6	0.024		[24,219]
		50% reduction (0.02 mM)		[212]
Maprotiline	0.95	0.028		[24,219]
Trazodone	ca. 1.0	20% reduction (0.1 mM)		[24,212]

Here, when IC_50_ values are not available, it is partly shown for comparison how channel current amplitudes are reduced by drugs, where % value indicates the extent of the reduction at the concentration shown in parentheses.

**Table 6 pharmaceuticals-13-00062-t006:** Comparison of IC_50_ values in inhibiting frog or rat sciatic nerve fast-conducting CAPs and TTX-sensitive or -resistant Na^+^ channels among local anesthetics.

Local Anesthetics	Frog CAP IC_50_ (mM)	Rat CAP IC_50_ (mM)	TTX-Sensitive Na^+^ Channel Current IC_50_ (mM)	TTX-Resistant Na^+^ Channel Current IC_50_ (mM)	References
Amide Type					
Lidocaine	0.74	0.28	0.204	0.073	[19,71,148,236]
Ropivacaine	0.34	0.28			[19,236]
Prilocaine	1.8				[26]
Levobupivacaine	0.23		0.264		[23,239]
Bupivacaine	76% reduction (0.5 mM)		0.027		[23,71]
			0.178		[155]
			0.190		[204]
Ester Type					
Cocaine	0.80	40% reduction (0.375 mM)		ca. 70% reduction (0.05 mM)	[20,72,244]
Procaine	2.2	ca. 1 mM	0.060		[27,71,248]
Benzocaine	0.80	37% reduction (1.3 mM)			[25,59]
Tetracaine	0.014		0.0007		[22,71]
Other Type					
Pramoxine	0.21				[26]

Here, when IC_50_ values are not available, it is partly shown for comparison how CAP and channel current amplitudes are reduced by drugs, where % value indicates the extent of the reduction at the concentration shown in parentheses.

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
