# Peer review of "Inhibition of Fast Nerve Conduction Produced by Analgesics and Analgesic Adjuvants—Possible Involvement in Pain Alleviation"

_pharmaceuticals, 2020, doi:10.3390/ph13040062_

Round 1

Reviewer 1 Report

Overall, this is an interesting review-article which summarize research focused in the inhibitory actions of the anti-nociceptive compounds on peripheral nerve conduction. The paper is well written and structured. In the paper, the anti-nociceptive compounds are grouped in narcotic, antipyretic, analgesic adjuvants, antiepilectics, antidepressants and local anesthetics. Usually the author gives examples of the anti-nociceptive action of each drug and then summary the research showing that the anti-nociceptive drug blocks or decreases the peripheral nerve conduction and them discuss the possible mechanism of nerve inhibition.

Although, the research summarized in the manuscript may influence the pain field, there is a key point that dampen my enthusiasm for the manuscript.

The sentence in line 22-23: “Nerve AP conduction inhibition produced by analgesics and analgesic adjuvants is suggested to contribute to at least a part of their antinociceptive effects ” works as the general idea that the manuscript wants to transmit to the readers.

However, all the research summarized in the manuscript focused in peripheral nerve conduction is on fast-conducting axons that do not transmit information related to pain.

With the idea to help to improve the impact of the manuscript I suggest taking in consideration the next,

  • Include research which the antinociceptive drugs are blocking slow-conducting Ad and C fibers that conduct nociceptive information.

Minor points:

  • Through the article the citations of the original papers and the reviews are not well differentiated
  • Line 212 there is a typo “loparamide” instead loperamide

Author Response

Reply to comments:

Thank you very much for reviewing my manuscript and giving me useful comments to revise this. I would like to reply to your comments as follows:

Overall, this is an interesting review-article which summarize research focused in the inhibitory actions of the anti-nociceptive compounds on peripheral nerve conduction. The paper is well written and structured. In the paper, the anti-nociceptive compounds are grouped in narcotic, antipyretic, analgesic adjuvants, antiepilectics, antidepressants and local anesthetics. Usually the author gives examples of the anti-nociceptive action of each drug and then summary the research showing that the anti-nociceptive drug blocks or decreases the peripheral nerve conduction and them discuss the possible mechanism of nerve inhibition.

Although, the research summarized in the manuscript may influence the pain field, there is a key point that dampen my enthusiasm for the manuscript.

The sentence in line 22-23: “Nerve AP conduction inhibition produced by analgesics and analgesic adjuvants is suggested to contribute to at least a part of their antinociceptive effects ” works as the general idea that the manuscript wants to transmit to the readers.

However, all the research summarized in the manuscript focused in peripheral nerve conduction is on fast-conducting axons that do not transmit information related to pain.

With the idea to help to improve the impact of the manuscript I suggest taking in consideration the next,

  • Include research which the antinociceptive drugs are blocking slow-conducting Ad and C fibers that conduct nociceptive information.

Response:

When available, I have added data about the effects of antinociceptive drugs on slow-conducting fiber CAPs; please see lines 941-946 in the revised manuscript.

Minor points:

  • Through the article the citations of the original papers and the reviews are not well differentiated

Response:

Throughout the text, I differentiated the reviews from the original papers; please see many places shown by red fonts in the revised manuscript.

  • Line 212 there is a typo “loparamide” instead loperamide

Response:

This mistype has been corrected; please see line 219 in the revised manuscript.

Reviewer 2 Report

This is an intensive review with many references of drug effects on peripheral nerve conduction of action potentials, mainly studied with frog sciatic nerve. In this review inhibitory actions on conduction of fast conducting fibers of many drugs studied mainly on frog sciatic nerve are compared. The author is eager to combine the results with analgesic effects of these drugs, although effects on slowly conducting thin fibers that are responsible for pain/nociception were not possible to examine. Conduction block of action potentials of large fibers will lead to muscle paralysis in case of muscle nerves, and it would be undesirable side effect. In several places the author forgets that AP conduction inhibitory effect of a drug is observed in fast conduction fibers, yet he discusses on analgesic effect of that drug. That is, there is a discrepancy between the title (and also the intension) of the paper and what is written. This is misleading, The author must be more cautious.  It would be necessary to discuss the gap between these two and possibility to expand the mechanism revealed in thick fibers to that in thin fibers, if such exist. For example, in case of local anesthetics such as lidocaine, blocking effect on Na channels is common to thick and thin fibers, and the order being blocked (thin fibers the first, thick fibers the last) is determined depending on the surface to volume ratio.

Another point to be considered is the concentration of drug that is effective in inhibiting conduction of action potentials in frog sciatic nerve and the local concentration when it is used in clinical setting. Comparison of these values would be helpful when one think whether such effect can occur in clinical condition.

Table 1 should be cited in more places so that the context of the paragraph may be claryfied. Or, Table 1 can be divided to several tables summarizing the effects of drugs in each group, and shown in the corresponding paragraph. I would recommend the latter.

Author Response

Thank you very much for reviewing my manuscript and giving me useful comments to revise this. I would like to reply to your comments as follows:

This is an intensive review with many references of drug effects on peripheral nerve conduction of action potentials, mainly studied with frog sciatic nerve. In this review inhibitory actions on conduction of fast conducting fibers of many drugs studied mainly on frog sciatic nerve are compared. The author is eager to combine the results with analgesic effects of these drugs, although effects on slowly conducting thin fibers that are responsible for pain/nociception were not possible to examine. Conduction block of action potentials of large fibers will lead to muscle paralysis in case of muscle nerves, and it would be undesirable side effect. In several places the author forgets that AP conduction inhibitory effect of a drug is observed in fast conduction fibers, yet he discusses on analgesic effect of that drug. That is, there is a discrepancy between the title (and also the intension) of the paper and what is written. This is misleading. The author must be more cautious.

Response:

Thank you very much for pointing out a critical issue regarding my manuscript. In order to decrease the extent of the discrepancy between the title of my paper and what is written, I have changed this title to “Inhibition of Fast Nerve Conduction Produced by Analgesics and Analgesic Adjuvants – Possible Involvement in Pain Alleviation”. Please see lines 2-4 in the revised manuscript.

The idea that conduction block of action potentials of large fibers will lead to muscle paralysis in case of muscle nerves, i.e., undesirable side effect, has been added to lines 954-955 in the revised manuscript.

It would be necessary to discuss the gap between these two and possibility to expand the mechanism revealed in thick fibers to that in thin fibers, if such exist. For example, in case of local anesthetics such as lidocaine, blocking effect on Na channels is common to thick and thin fibers, and the order being blocked (thin fibers the first, thick fibers the last) is determined depending on the surface to volume ratio.

Response:

Thank you very much for your thoughtful comment. These ideas also have been added to lines 955-958 in the revised manuscript.

Another point to be considered is the concentration of drug that is effective in inhibiting conduction of action potentials in frog sciatic nerve and the local concentration when it is used in clinical setting. Comparison of these values would be helpful when one think whether such effect can occur in clinical condition.

Response:

When available, I have given clinical effective concentrations of the drugs used, i.e., see line 142 for tramadol, line 456 for DEX, lines 580-582 for antiepileptics and lines 672-676 for antidepressants in the revised manuscript. These concentrations are much lower than those at which frog sciatic nerve CAPs are inhibited. Therefore, I think that nerve conduction inhibition may occur when the drugs is administrated locally or is accumulated in the nervous system. This idea has been mentioned in lines 951-954 in the revised manuscript.

Table 1 should be cited in more places so that the context of the paragraph may be claryfied. Or, Table 1 can be divided to several tables summarizing the effects of drugs in each group, and shown in the corresponding paragraph. I would recommend the latter.

Response:

Thank you very much for your invaluable instruction. According to your latter recommendation, Table 1 was divided to five tables (please see pages 5, 7-8, 9-10, 12, 13-14 and 15-16 in the revised manuscript) and each of them was shown in the corresponding paragraph.

Reviewer 3 Report

Dear Author,

I would like to express my sincere compliments for this review article you wrote. This article is well written, is complete and is very useful to all people involved in both in pain research and in pain clinics.

Author Response

Thank you very much for your reviewing my manuscript.

Reviewer 4 Report

This review presents inhibition of nerve conduction by analgesics and analgesic adjuvants.  The review provides a helpful contribution to action potential conduction in nerve fibers and transmission of nociceptive information.  However, there are a few points that I would like to highlight:

  1. ll.91-94

Opioids, fentanyl and sufentanil, inhibit sciatic nerve CAP amplitude and AP conduction in frog sciatic nerve and mammalian peripheral nerves.  However, there is some kind of difference in the sensitivity of naloxone.  What does the difference depend on ?    

  1. ll.123-124

Please add tramadol in the sentence:

the CAP inhibition → the tramadol-induced CAP inhibition

  1. ll.145-148, 155-158, 164-167, 295-304, 435-437

CAP amplitude reduction increased in extent with an increase in the number of -CH2.  The discussion is very important for understanding of target structure.  Further, there are relationship between benzene rings and the CAP inhibition.  Please add figures of chemical structure concerning drugs to further understand the chemical structure-specific CAP amplitude reduction.

  1. ll.314-318

Among NSAIDs, high concentration of NSAIDs such as diclofenac blocks the CAP amplitude via probably the inhibition of voltage-gated Na+ channels.  Other NSAIDs such as aspirin have no effects on the CAP amplitude.  Taken together, prostaglandins do not affect the CAP amplitude of nerve fibers directly.  Please add the discussion concerning effect of prostaglandins. 

  1. ll.611-652

The comparison in the efficiency of nerve conduction inhibition among analgesics and analgesic adjuvants is very important.  However, you focused only IC50 vales.  Please add the comparison between IC50 and drug structures.

Author Response

Thank you very much for reviewing my manuscript and giving me useful comments to revise this. I would like to reply to your comments as follows:

This review presents inhibition of nerve conduction by analgesics and analgesic adjuvants.  The review provides a helpful contribution to action potential conduction in nerve fibers and transmission of nociceptive information.  However, there are a few points that I would like to highlight:

  1. ll.91-94

Opioids, fentanyl and sufentanil, inhibit sciatic nerve CAP amplitude and AP conduction in frog sciatic nerve and mammalian peripheral nerves.  However, there is some kind of difference in the sensitivity of naloxone.  What does the difference depend on ?

Response:

Although fentanyl and sufentanil reduce CAP peak amplitudes in the mammalian (rabbit) peripheral (vagus) nerve, these reductions are seen in the presence of naloxone (Gissen et al., 1987; [39]). The extents of the reductions by fentanyl and sufentanil in the presence of naloxone appear to be comparable to those in the absence of naloxone, as judged from a comparison of Fig. 1 (right) and Fig. 3 (for fentanyl) and also of Fig. 2 (right) and Fig. 4 (for sufentanil) (Gissen et al., 1987; [39]) and thus there does not appear to be a difference in naloxone sensitivity between fentanyl and sufentanil activities. As reported by Haeseler et al. (2006; [65]), the CAP inhibitions produced by fentanyl and sufentanil are possibly mediated by their inhibitory actions on voltage-gated Na+ channels.

  1. ll.123-124

Please add tramadol in the sentence:

the CAP inhibition → the tramadol-induced CAP inhibition

Response:

I have added “tramadol-induced”; please see line 125 in the revised manuscript.

  1. ll.145-148, 155-158, 164-167, 295-304, 435-437

CAP amplitude reduction increased in extent with an increase in the number of -CH2.  The discussion is very important for understanding of target structure.  Further, there are relationship between benzene rings and the CAP inhibition.  Please add figures of chemical structure concerning drugs to further understand the chemical structure-specific CAP amplitude reduction.

Response:

We had already shown in our previous papers the chemical structures of the opioids, NSAIDs and antiepileptics tested. Therefore, I have given the numbers of figures illustrating their chemical structures in the published papers. For ll.145-148, (155-158, 164-167), 295-304 and 435-437 pointed out by you, please see lines 151-152, 169, (328 and 333-334) and 548, respectively, in the revised manuscript. Moreover, for the chemical structures of antidepressants and local anesthetics used, I also have referred the figure numbers of our previous papers; please see line 662 and 788-789 in the revised manuscript.

  1. ll.314-318

Among NSAIDs, high concentration of NSAIDs such as diclofenac blocks the CAP amplitude via probably the inhibition of voltage-gated Na+ channels.  Other NSAIDs such as aspirin have no effects on the CAP amplitude.  Taken together, prostaglandins do not affect the CAP amplitude of nerve fibers directly.  Please add the discussion concerning effect of prostaglandins.

Response:

I do not know why the inhibitory effect of diclofenac and no effect of aspirin on Na+ channels are related to no direct effect of prostaglandins on the channels. There appears to be a difference among different types of neuron in the modulation of action potentials produced by prostaglandin E2 (Gold & Traub, Cutaneous and colonic rat DRG neurons differ with respect to both baseline and PGE2-induced changes in passive and active electrophysiological properties; J. Neurophysiol. 2004, 91, 2524-2531).

  1. ll.611-652

The comparison in the efficiency of nerve conduction inhibition among analgesics and analgesic adjuvants is very important.  However, you focused only IC50 vales.  Please add the comparison between IC50 and drug structures.

Response:

Although there was a relationship between CAP inhibition extent and the number of CH2 in opioids having similar structures, as mentioned in Section 2.1, I did not note a chemical structure common among analgesic adjuvants having IC50 values of 0.2-0.5 mM or 1-2 mM. Furthermore, no common chemical structure was found between NSAIDs and analgesic adjuvants having similar IC50 values. These facts have been mentioned in lines 832-835 and 847-848 in the revised manuscript.

Round 2

Reviewer 1 Report

All my concerns were answered.

Author Response

Response:

Thank you very much for your reviewing my revised manuscript.

Reviewer 2 Report

The manuscript has been well revised according/answering to my comments. One point to be considered is that tables newly made are not cited or used in the text. The author can use these tables to make the text clearer.

Author Response

Thank you very much for your reviewing my revised manuscript. According to your instruction, I cited newly-made Tables 1-6 on pages 16 and 17 in the re-revised manuscript.